



# Database of Petrophysical Properties of the Mid-German Crystalline High

Sebastian Weinert[1,2], Kristian Bär[1], Ingo Sass[1,2]

[1]Geothermal Science and Technology, Technical University of Darmstadt, Darmstadt, 64287, Germany
5 [2]Darmstadt Graduate School of Excellence Energy Science and Engineering, Technical University of Darmstadt, Darmstadt, 64287, Germany

*Correspondence to*: Sebastian Weinert (weinert@geo.tu-darmstadt.de)

**Abstract.** Petrophysical properties are a key element for reservoir characterization but also for interpreting the results of various geophysical exploration methods or geophysical well logs. Furthermore, petrophysical properties are commonly used 10 to populate numerical models and are often critically governing the model results. Despite the common need of detailed petrophysical properties, data is still very scarce and often not available for the area of interest. Furthermore, both the online research for published property measurements or compilations, as well as dedicated measurements campaigns of the selected properties, which requires comprehensive laboratory equipment, can be very time-consuming and costly. To date, most published research results are often focused on a limited selection of parameters only and hence, researching various 15 petrophysical properties, needed to account for the thermal-hydraulic-mechanical behavior of selected rock types or reservoir settings, can be very laborious.

Since for deep geothermal energy in central Europe, the majority of the geothermal potential or resource is assigned to the crystalline basement, a comprehensive database of petrophysical properties comprising rock densities, porosity, rock matrix permeability, thermal properties (thermal conductivity and diffusivity, specific heat capacity) as well as rock mechanical 20 properties as compressive and shear wave velocities, unconfined compressive strength, Young's modulus, Poisson's ratio, tensile strength and triaxial shear strength was compiled by measurements conducted at the HydroThermikum lab facilities of the Technical University of Darmstadt.

Analyzed samples were mostly derived from abandoned or active quarries and natural or artificial outcrops such as road cuts, river banks or steep hill slopes. Furthermore, samples of the cored deep wells Worms 3 (samples from 2175-2195 m), 25 Stockstadt 33R (samples from 2245-2267 m), Weiterstadt 1 (samples from 2502-2504 m), Tiefbohrung Groß-Umstadt/Heubach, B/89 – B02 and the cored shallow wells Forschungsbohrung Messel GA 1 and 2 as well as GWM17 Zwingenberg, GWM1A Zwingenberg, Langenthal BK2/05, EWS267/1 Heubach, and archive samples of the Institut für Steinkonservierung e. V. in Mainz originating from a comprehensive large scale sampling campaign in 2007 were investigated. The database (Weinert et al. 2020b, https://doi.org/10.25534/tudatalib-278) aims to provide easily accessible petrophysical 30 properties of the Mid-German Crystalline High, measured on 224 locations in Bavaria, Hesse, Rhineland-Palatinate and





Thuringia and comprising 26,951 single data points. Each data point is addressed with the respective metadata such as sample identifier, sampling location, petrography and if applicable stratigraphy and sampling depth (in case of well samples).

## 1 Introduction

For geothermal energy, reservoir exploration often lacks dedicated slim-hole exploration wells to enhance the understanding of the physical and hydraulic behavior of the explored geothermal reservoir at an early project stage (Sass et al. 2016). Therefore, reservoir characterization often solely relies on either geophysical exploration or numerical models which, in turn, need petrophysical input parameters to be successful and accurate. Due to the sparseness of reservoir samples, explained by the high costs of coring deep wells, petrophysical properties can be derived from outcrop analogue studies (e.g. Howell et al., 2014 or Ukar et al., 2019) or literature data for suitable rock types (Bär et al., 2020). Nonetheless, even in rather isotropic, homogeneous material such as crystalline rocks, petrophysical properties can vary depending on their geochemical composition and texture but also physical appearance, micro-fractures or porosity as well as degree of alteration or weathering. A profound understanding of the regarded rock type, its respective petrophysical properties and the methods how those were measured are essential for populating numerical simulations or interpreting geophysical exploration methods.

Despite the importance of petrophysical properties, as well as the importance of crystalline basement rocks in deep geothermal energy, to which 85% to 90% of the German geothermal potential is accredited (Deutscher Bundestag, 2003), such data is often either unpublished, only published for confined areas (e. g. Mielke et al., 2016, Aretz et al., 2016, Weydt et al., 2020, Weinert et al., 2020a) or published without important meta-information. The search for suitable petrophysical properties can therefore be very time-consuming and often, only widely averaged properties can be found and are commonly used neglecting local heterogeneities, vertical and lateral variability and anisotropic behavior of the rocks.

To overcome the lack of suitable petrophysical data, a sampling and measuring campaign was initiated within the scope of the Hessen 3D 2.0 project: '3D-Modell der petrothermalen und mitteltiefen Potenziale zur Wärmenutzung und -speicherung von Hessen' (Federal Ministry for Economic Affairs and Energy; funding number 0325944A) with the aim to develop a comprehensive database. This database of petrophysical properties of the Mid-German Crystalline High was supplemented and compiled to facilitate easy access to research data on measured petrophysical properties and to allow for an adequate generalization for specific petrological units within the Mid-German Crystalline High. Therefore, the database presented here (Weinert et al. 2020b, https://doi.org/10.25534/tudatalib-278) publicly provides all relevant laboratory measurements on the Mid-German Crystalline High samples of a variety of unpublished and published studies of the Technical University of Darmstadt as well as over 1,900 newly measured data points.



## 2 Mid-German Crystalline High

The Mid-German Crystalline High (MGCH) is a NE-SW striking Variscan complex of approx. 50-65 km width in NW-SE extension and several hundred kilometers length along strike. Locally exposed in the Pfälzer Wald, Odenwald, Spessart, Ruhla Mountains and Kyffhäuser Crystalline Complex, the MGCH is sandwiched between the Saxothuringian Zone to the SW and the Northern Phyllite Zone, which represents the southern suture zone of the Rhenohercynian belt to the NE (Fig. 1). While the NW boundary between the MGCH and Northern Phyllite Zone is not exposed, the MGCH is fault-bounded to the

Saxothuringian zone (Linnemann et al., 2008) to the SE.

The MGCH metamorphic and crystalline complexes are interpreted as the northern margin of Armorica (McCann et al., 2008) and hence a suture of the Rheic Ocean at the rim of the Bohemian Massif (Linnemann et al., 2008).

Outcrops of the MGCH display a variety of high-grade metamorphic Late Ordovician to Early Devonian rocks in the Böllsteiner Odenwald (450 Ma, Reischmann et al., 2001), the Spessart Crystalline Complex (418-407 Ma, Lippolt, 1986,

Dombrowsky et al., 1995) or the Ruhla Crystalline Complex (413-400 Ma, Brätz, 2000, Zeh and Wunderlich, 2003). Mafic, intermediate and acid intrusive igneous rocks are preferentially exposed in the Bergsträsser Odenwald where they comprise up to 90% of the exposed rocks (Stein, 2001), as well as in the Spessart and Ruhla mountains. The northern part of the Odenwald (Frankenstein Complex) is predominantly comprised of Late Devonian gabbro (362±7 Ma, Kirsch et al., 1988) as well as metamorphic rocks. The southern part is composed of amphibolite-facies metamorphosed metasediments and gneiss

(342-332 Ma, Todt et al., 1995) which were intruded by the undeformed Weschnitz, Tromm and Heidelberg plutons. Those intrusions are homogenous and comprised by monzodiorite to granodiorite (Weschnitz pluton), granite (Tromm pluton) and gabbro to diorite with later granite and granodiorite intrusions at the Heidelberg pluton (Timmerman, 2008). Post-tectonic carboniferous diorite and granodiorite dominate the SE part of the Spessart Crystalline complex (c. 330 Ma, Anthes and Reischmann, 2001) while Carboniferous granites predominantly occur in the Ruhla mountains (Timmerman, 2008).




**Figure 1: Simplified overview map after (Hirschmann 1995, Voges et al. 1993, Klügel 1997, Kroner et al. 2008) of the Mid-German Crystalline High outcrops (A) Odenwald (after Stein 2001; Will and Schmädicke 2001; McCann et al. 2008), (B) Spessart (after Okrusch 1983; Dombrowski et al. 1995; McCann et al. 2008) and (C) Ruhla Mountains (Zeh et al. 2003; McCann et al. 2008).**






## 2 Contents and Structure of the Database

The database is provided in spreadsheet format as well as in delimited text file format. It is structured in two super-entities, namely 'metadata' and 'petrophysical properties' and further hierarchical structured into logical subdivisions.

While the metadata stores information about the sample location, sample ID but also stratigraphic and petrographic
information, the actual measured petrophysical properties are summarized under petrophysical properties.

Each super-entity and its content are described in following sub-chapters in detail.

### 2.1 Metadata

The super-entity 'Metadata' comprises information concerning the sample identifier (sample ID) and its parent ID, the analyzer of the petrophysical properties, the sampling location but also stratigraphic and petrographic information and the sample
dimensions if measured and documented.

#### 2.1.1 Sample Information

The presented database renounces to unify sample IDs as other researches did (e.g. Bär et al., 2020) and instead only documents the original sample ID that was chosen by the analyzer. If multiple measurements were performed on a single sample (e.g. thermal conductivity on top and bottom end faces of cylindrical samples) the parental sample ID of the actual specimen is also
provided. Therefore, reviewing original sources is feasible and allows to easily search for samples and further information such as detailed descriptions in the original sources. For all data extracted from either published or unpublished theses or reports, the referring reference is also given and indexed in the bibliography of both, the database (Weinert et al., 2020b) as well as the here presented work.

Although all the presented data were analyzed in the same institution applying the same laboratory equipment, the individual
person analyzing the samples might affect the data during data gathering and result evaluation (for example picking shear wave velocities). While comparing datasets of different people slight variations can occur. Therefore, the person who has analyzed the samples is also stored in the sample information.

#### 2.1.2 Sampling Location

Samples were taken from quarries, abandoned quarries, outcrops, wells (cored borehole sections) as well as the archive of the
Institut für Steinkonservierung e. V. Knowledge of the sample origin is important in data evaluation. For example, samples taken in active quarries might be influenced due to excavation either by heavy machinery or explosives while samples from abandoned quarries and natural outcrops may be slightly to significantly weathered. Well samples were subdued to higher





temperature and pressure conditions and might show extension (micro-)fractures due to stress relief during core retrieval ('core-disking').

In addition to the outcrop type, information about the geographical origin of the sample are given by a location as well as the referring federal state and state of this location (e.g. location: E of Wingertsstraße, Alzenau; federal state: Hesse; country: Germany). Also the location coordinates are indexed in the database and catalogued as decimal degree with the reference system WGS84. The elevation at the coordinate point is given in meters above mean sea level (MAMSL).

For well samples besides geographical information, also the well name and its respective archive number (if applicable) are

indexed. The given elevation correlates with the well head elevation and the depth of the sample conforms to the measured depth (MD) below ground level (b.g.l.).

### 2.1.3 Stratigraphy

The stratigraphy documented in the database is divided into the period and series representing the sample. Additionally, a term for the formation or unit is given, which, if given, conforms with the locally used term.

According to international standards, the documented terms used for the stratigraphic period and series are corresponding to the international chronostratigraphic chart of the IUGS v2020/01 (Cohen et al., 2013, updated). For each stratigraphic period and series, a respective stratigraphic ID is provided which correlates with the stratigraphic IDs published in Bär et al. (2019a) and Bär et al. (2020).

### 2.1.4 Petrography

Petrographic metadata are given by a simplified petrographic term and a correlating petrographic ID and petrographic parent ID (Fig. 2). The petrographic IDs are corresponding to Bär et al. (2019b) which are based on the well database of Hessian Agency for Nature Conservation, Environment and Geology. Also based on the petrographic ID presented in Bär et al. (2019b) a parental petrographic ID for each sample is provided.

### 2.1.4 Sample Dimensions

If measured, the sample height and diameter are reported in centimeter (cm). Additionally, the grain and bulk volume are documented in grams per cubic centimeter ($g \; cm^{-3}$) and sample weight is reported in grams (g).

### 2.2 Petrophysical Properties

The database presented here includes 20 kinds of petrophysical properties measured on a variety of samples. To increase readability, petrophysical properties are subdivided into thermophysical and rock mechanical properties (Table 1 and Table

2). All analysis performed on the same specimen are documented in the same row. For some methods such as thermal conductivity and diffusivity, multiple measurements at the identical sample are possible (e.g. top and bottom end face of the



sample). In case of multiple measurements, each single measurement is documented in a separate row (sample ID). Additionally, an average value and standard deviation is given for the specimen (parental sample ID).

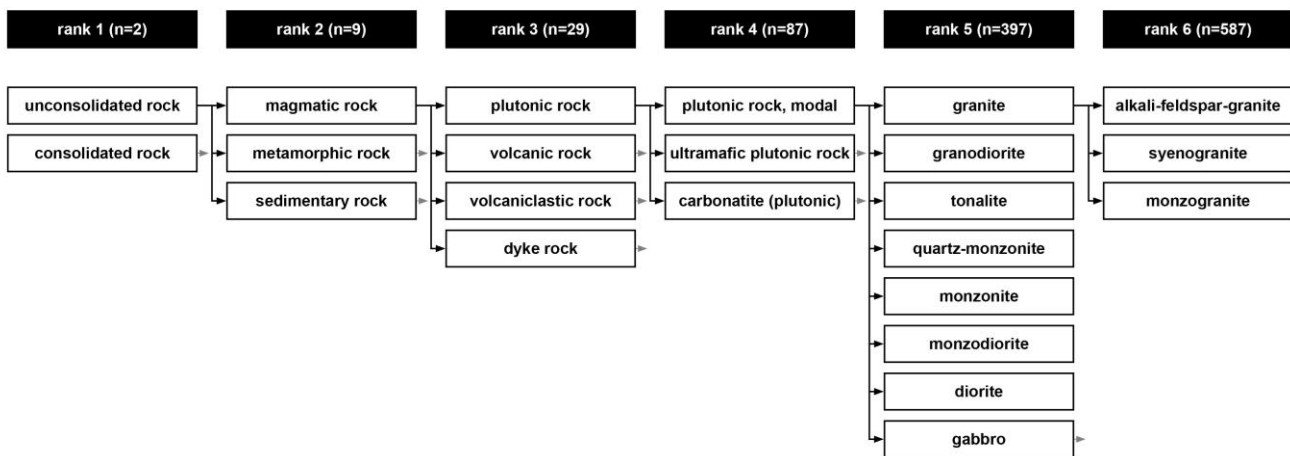


**Figure 2: Hierarchical system of standardized petrographic terms referring to Bär et al. 2019b. Black boxes show the rank and number of entries within each rank, white boxes represent the specific term and show classification scheme on the example of granites. Black arrows indicate direct connections and grey arrows represent additional terms not displayed here.**

### 2.2.1 Thermophysical Properties

Documented thermophysical properties include grain and bulk density, porosity, thermal conductivity and diffusivity, specific heat capacity but also apparent and intrinsic permeability. Since thermal conductivity and thermal diffusivity is often measured on multiple surfaces both parameters have three dedicated rows each. One row documents single measurements, one the average value of the sample and the last row documents the standard deviation of the average.

The total number of single thermophysical property measurements and average values are listed in Table 1.


**Table 1: Number of measurements of thermophysical properties.**

| Property | Number of single measurements | Number of measured samples |
|---|---|---|
| Bulk density | 974 | 974 |
| Grain density | 1,238 | 1,238 |
| Porosity | 918 | 918 |
| Thermal conductivity | 3,518 | 1,551 |
| Thermal diffusivity | 3,298 | 1,512 |
| Specific heat capacity | 1,415 | 1,415 |
| Apparent and intrinsic permeability | 991 | 991 |
| **Total** | **12,353** | **8,600** |



All measurements were conducted at the Technical University of Darmstadt and the applied methods are described in the following section.

**2.2.2 Rock Mechanical Properties**

Compressive and shear wave velocity, dynamic and static Young's modulus, dynamic and static Poisson's ratio, unconfined compressive strength, bulk modulus, tensile strength as well as triaxial shear strength, cohesion and angle of friction are summarized (Table 2). Compressive and shear wave velocities are often measured multiple times at the same specimen; therefore, each single measurement is listed as well as the respective sample average including the standard deviation. Since

dynamic Young's modulus and Poisson's ratio are calculated using the compressive and shear strength, a sample average and standard deviation is also provided for them.

Table 2: Number of measurements of rock mechanical properties.

| Property | Number of single measurements | Number of measured samples |
| --- | --- | --- |
| Compressive wave velocity | 1,247 | 822 |
| Shear wave velocity | 1,247 | 822 |
| Dynamic Young's modulus | 1,165 | 740 |
| Dynamic Poisson's Ratio | 1,247 | 822 |
| Unconfined compressive strength | 400 | 400 |
| Static Young's modulus | 185 | 185 |
| Static Poisson's Ratio | 180 | 180 |
| Bulk modulus | 116 | 116 |
| Tensile Strength | 231 | 231 |
| Triaxial Shear Strength | 106 | 106 |
| **Total** | **5,816** | **4,116** |

As for the petrophysical properties, all measurements were conducted at the Technical University of Darmstadt and the applied methods are described in the following section.

**3 Methods**

Measurements comprise grain density, bulk density, porosity, apparent and intrinsic permeability, thermal conductivity, thermal diffusivity, compressional and shear wave velocities as well as and at selected locations unconfined compressive



strength, Young's modulus, Poisson's ratio, tensile strength and triaxial shear strength including cohesion and coefficient of friction.

Prior to any measurement, samples were preferably cut to plugs of 40 mm diameter and 30 mm length and in case of rock mechanical tests to cores of a length to diameter ratio of 2:1 at a diameter of 64 and 55 mm and the core end faces were ground parallel and perpendicular to the core axis. Tensile strength analyses were performed on disks with a height to diameter ratio

of 1:2 at diameters of 64 mm and 55 mm. Although, some analyses were also performed on different sample dimensions if the method allows.

All measurements were conducted on oven-dry specimens at laboratory conditions of an average atmospheric pressure of about 0.1 MPa and at 20 °C for thermal rock properties and at approximately 23 °C for other petrophysical properties.

As follows, all applied methods are briefly described. For a more detailed methodological description please refer to Weydt et

al. (2020, in preparation).

### 3.1 Thermophysical Properties

Grain density was determined in a gas expansion pycnometer (AccuPyc II 1340) applying helium as displacement fluid. The accuracy for grain density measurements is stated by the manufacturer as 0.02 %. Each data point is sampled from 5 single measurements. Bulk density measurements are measured with an envelope density analyzer (GeoPyc 1360). A well sorted,

fine-grained powder (Dry Flo) is utilized as displacement material to determine the bulk volume of the specimen. Based on bulk volume and weight of the specimen, bulk density is calculated, which then, together with the grain density, is used for the calculation of the specimen's gas effective porosity. The accuracy is given by the manufacturer to be within 1.1 %. Measurements comprise 3 single measurements which have been are averaged. Negative porosity values are reported on very low porous samples, but should be seen as samples with >1.1% porosity.

Dry bulk thermal conductivity and thermal diffusivity was measured using a commercial thermal conductivity scanner (Lippmann and Rauen TCS) applying the optical scanning method after Popov et al. (1999). Both parameters are measured by temperature sensors mounted on a slide moving beneath a sampling area along the core axis. For the thermal conductivity, samples and a pair of references are heated up by approximately 4°C in comparison to the ambient temperature. Thermal diffusivity is determined almost equally except that the temperature is additionally measured with a third temperature sensor

shifted 7 mm perpendicular to the axis of movement. During scanning, every 1 mm a single measuring point is logged and at each logging point thermal diffusivity and conductivity is calculated based on the subsequent heating rate. The manufacturer states an accuracy within 3% of the thermal conductivity and 5% of the thermal diffusivity measurement.

Specific heat capacities are calculated after Buntebarth (1984)

$$c_p = \frac{\lambda}{\rho_{bulk} \cdot \kappa} \tag{1}$$

where $c_p$ is the specific heat capacity, $\lambda$ is the thermal conductivity, $\rho_{bulk}$ is the bulk density and $\kappa$ the thermal diffusivity.



The intrinsic permeability was measured based on the principle of Klinkenberg (1941) using a column gas permeameter, which measures the permeability of plugs applying a pressure gradient between the top and bottom surface of the sample mounted in a Hassler-cell (Filomena et al. 2014). Since the intrinsic permeability reflects the effective gas permeability under infinite pressure, the intrinsic permeability is extrapolated based on the Klinkenberg plot of multiple single measurements of different injection pressures at constant pressure gradients.

Apparent permeability is measured with a mini permeameter, which measures the permeability at various injection rates in the nearfield of a nozzle, pressed against the sample surface. Like the column permeameter, the mini permeameter utilizes air as measuring medium.

## 3.2 Rock Mechanical Properties

Ultrasound wave velocities were measured using the commercial ultrasound generator Geotron USG 40 with mounted UPE-S (receiver) and UPG-S (emitter) probes, which enhance the shear wave signature. Both probes are pressed against the center of each end surface of the specimen. The contact pressure was set to 0.1 MPa and a shear gel (Magnaflux 54-T04) was applied between sample and probe to enhance the transmission of shear waves.

Dynamic elastic parameters are calculated with the equations

$$v_{dyn} = \frac{V_p^2 - 2V_s^2}{2(V_p^2 - V_s^2)} \tag{2}$$

$$E_{dyn} = \rho_{bulk} v_p^2 \frac{(3V_p^2 - 4V_s^2)}{(V_p^2 - V_s^2)} \tag{3}$$

where $v_{dyn}$ is the dynamic Poisson coefficient, $E_{dyn}$ is the dynamic Young's modulus, $\rho_{bulk}$ is the bulk density and $V_p$ and $V_s$ are the compressional and shear wave velocity.

Measurements were averaged out of 16 single measurements with a frequency of 80 kHz or 250 kHz, depending on the sample dimension and shape.

Unconfined compressive strength was force as well as displacement controlled tested in a 1000 kN test frame (Form+Test Prüfsysteme) according to the ASTM D7012.

Elastic properties (Young's modulus and Poisson's ratio) are also measured in a 1000 kN test frame (Form+Test Prüfsysteme) according to the ASTM D3148 and Mutschler (2004).

Triaxial shear strength are measured in a 1000 kN test frame (Wille Geotechnik) on samples of 55 mm in diameter and 110 mm in length according to the ASTM D2664. Confining pressures up to 30 MPa were chosen, although for hard rocks commonly 5, 10 and 20 MPa are applied.

Tensile strength is determined on rock disks of 55 and 64 mm diameter at length ratios of approx. 0.5:1 (diameter to length according to ASTM D3967-16.)





## 4 Status of the Database and Quality

To date, the database is comprised by data of either student's theses, scientific reports or self-supervised measurements conducted at the Technical University of Darmstadt. In total, 5,204 data rows (Table 3) are entered in the database which are distributed over 224 locations including quarries, abandoned quarries, outcrops and wells.

**Table 3: Quantity of data entries lines categorized after the petrographic ID (rank 3, Bär et al. 2019b).**

| Petrographic ID | 10105 | 10210 | 10322 | 10881 | 10907 | 69619 |
|---|---|---|---|---|---|---|
| Petrographic Term | Plutonic Rock | Volcanic Rock | Dyke rock | Metamorphic Rock (after educt) | Metamorphic Rock (after chemistry, fabric, mineral content) | Tectonite |
| Data rows documented | 3652 | 38 | 170 | 60 | 1216 | 68 |

As shown in Figure 3 the chosen sample locations are spread across most German outcrops of the MGCH, although many of the sampling points are concentrated in the Crystalline Odenwald. Data of the Kyffhäuser Crystalline Complex are not included in the database so far. Also, data from further wells from either the Bavarian or Rhineland-Palatinate geological services might complement the database in the future. As shown in Figure 3, more data from the Ruhla Mountains and Spessart is desirable while the Odenwald is densely sampled. Nonetheless, more complex units, such as the Flasergranitoid Zone in the northern Odenwald, should also be sampled more densely and in future, the scope should be extended to include bulk rock geochemical analysis.

Unlike databases that compile literature data of petrophysical properties (e.g. Bär et al. 2020), all data in the presented database were measured at the same institute. Therefore, the dataset is homogeneous and all data were measured under equivalent conditions and application of the same laboratory devices. Furthermore, all measurements were performed by qualified lab professionals. Quality control was therefore an ongoing process while supervising the analyzers and reviewing the final raw and processed data, resulting in a homogeneous dataset. Quality control and reliability indices as for example presented in Bär et al. (2020) are not necessary in the presented dataset.




**Figure 3: Sampling location of the presented database scattered around the Odenwald, Spessart and Ruhla mountains as well as well locations and outcrops in the Pfälzer Wald. Simplified overview map after (Hirschmann 1995, Voges et al. 1993, Klügel 1997, Kroner et al. 2008) of the Mid-German Crystalline High outcrops. Sampling locations from Bär (2008), Bär (2012), Biewer (2017), Dutheillet de Lamothe (2016), Hoffmann (2015), Klaeske (2010), Lambert (2016), Maire (2014), Pei (2009), Rüther (2011), Schäffer**
**(2012), Schäffer et al. (2018), Vogel (2016), Weber (2014), Welsch (2012) and Welsch et al. (2015).**



## 4.1 Data Evaluation

Table 3 comprises minimum and maximum data of bulk density, gran density, porosity, thermal conductivity, thermal diffusivity as well as compressive and shear wave velocity of all samples reported in Weinert et al. (2020b). Besides the
minimum and maximum values, an average value for each petrographic ID as well as parent ID is calculated and reported with a referring standard deviation and the number of samples (n). Figure 4 displays the sample data in cross scatter plots.

**Table 4: Petrophysical Properties averaged over the measured samples for each petrographic ID given in the Database Weinert et al. (2020b).**

| | Min | Max | Average | Standard Deviation | n |
|---|---|---|---|---|---|
| **10105 – Plutonic Rock: Total Average** | | | | | |
| Bulk Density [g cm$^{-3}$] | 2.35 | 2.99 | 2.68 | 0.10 | 671 |
| Grain Density [g cm$^{-3}$] | 2.35 | 3.08 | 2.74 | 0.10 | 866 |
| Porosity [%] | -1.43 | 23.79 | 1.89 | 1.97 | 632 |
| Thermal Conductivity [W m$^{-1}$ K$^{-1}$] | 0.35 | 4.22 | 2.44 | 0.43 | 1066 |
| Thermal Diffusivity [x10$^{-6}$ m$^2$ s$^{-1}$] | 0.53 | 2.63 | 1.22 | 0.26 | 1048 |
| Compressive Wave Velocity [m s$^{-1}$] | 1776 | 8167 | 4795 | 1116 | 631 |
| Shear Wave Velocity [m s$^{-1}$] | 1027 | 4811 | 2680 | 659 | 631 |
| **10110 – Granite** | | | | | |
| Bulk Density [g cm$^{-3}$] | 2.39 | 2.95 | 2.62 | 0.08 | 238 |
| Grain Density [g cm$^{-3}$] | 2.57 | 3.00 | 2.67 | 0.07 | 274 |
| Porosity [%] | -1.43 | 9.55 | 1.93 | 1.59 | 233 |
| Thermal Conductivity [W m$^{-1}$ K$^{-1}$] | 1.73 | 4.22 | 2.74 | 0.42 | 293 |
| Thermal Diffusivity [x10$^{-6}$ m$^2$ s$^{-1}$] | 0.76 | 2.63 | 1.44 | 0.28 | 292 |
| Compressive Wave Velocity [m s$^{-1}$] | 1776 | 7208 | 4711 | 1116 | 225 |
| Shear Wave Velocity [m s$^{-1}$] | 1100 | 4038 | 2623 | 679 | 225 |






**Continuation of Table 3: Petrophysical Properties averaged over the measured samples for each petrographic ID given in the Database Weinert et al. (2020b).**

| 10114 – Granodiorite | | | | | |
|---|---|---|---|---|---|
| Bulk Density [g cm$^{-3}$] | 2.42 | 2.87 | 2.69 | 0.07 | 296 |
| Grain Density [g cm$^{-3}$] | 2.35 | 2.87 | 2.73 | 0.07 | 378 |
| Porosity [%] | 0.01 | 12.27 | 1.82 | 1.88 | 262 |
| Thermal Conductivity [W m$^{-1}$ K$^{-1}$] | 0.35 | 3.39 | 2.48 | 0.36 | 394 |
| Thermal Diffusivity [x10$^{-6}$ m$^2$ s$^{-1}$] | 0.72 | 2.08 | 1.22 | 0.19 | 386 |
| Compressive Wave Velocity [m s$^{-1}$] | 1802 | 7046 | 4489 | 975 | 284 |
| Shear Wave Velocity [m s$^{-1}$] | 1027 | 3984 | 2541 | 561 | 284 |
| **10115 – Tonalite** | | | | | |
| Bulk Density [g cm$^{-3}$] | - | - | - | - | - |
| Grain Density [g cm$^{-3}$] | - | - | - | - | - |
| Porosity [%] | - | - | - | - | - |
| Thermal Conductivity [W m$^{-1}$ K$^{-1}$] | 2.60 | 2.66 | 2.63 | 0.04 | 2 |
| Thermal Diffusivity [x10$^{-6}$ m$^2$ s$^{-1}$] | 1.15 | 1.15 | 1.15 | 0.00 | 2 |
| Compressive Wave Velocity [m s$^{-1}$] | - | - | - | - | - |
| Shear Wave Velocity [m s$^{-1}$] | - | - | - | - | - |
| **10122 – Quartz Diorite** | | | | | |
| Bulk Density [g cm$^{-3}$] | 2.58 | 2.78 | 2.73 | 0.04 | 27 |
| Grain Density [g cm$^{-3}$] | 2.78 | 2.99 | 2.81 | 0.04 | 38 |
| Porosity [%] | 0.65 | 8 | 2.27 | 1.47 | 27 |
| Thermal Conductivity [W m$^{-1}$ K$^{-1}$] | 1.93 | 2.57 | 2.22 | 0.16 | 38 |
| Thermal Diffusivity [x10$^{-6}$ m$^2$ s$^{-1}$] | 0.91 | 1.39 | 1.07 | 0.10 | 38 |
| Compressive Wave Velocity [m s$^{-1}$] | 4172 | 6780 | 5674 | 642 | 38 |
| Shear Wave Velocity [m s$^{-1}$] | 1883 | 3979 | 3080 | 458 | 38 |




**Continuation of Table 3: Petrophysical Properties averaged over the measured samples for each petrographic ID given in the Database Weinert et al. (2020b).**

| 10123 – Quartz Gabbro | | | | | |
|---|---|---|---|---|---|
| Bulk Density [g cm$^{-3}$] | - | - | - | - | - |
| Grain Density [g cm$^{-3}$] | 2.84 | 2.84 | 2.84 | - | 1 |
| Porosity [%] | - | - | - | - | - |
| Thermal Conductivity [W m$^{-1}$ K$^{-1}$] | 2.35 | 2.35 | 2.35 | - | 1 |
| Thermal Diffusivity [x10$^{-6}$ m$^2$ s$^{-1}$] | 1.25 | 1.25 | 1.25 | - | 1 |
| Compressive Wave Velocity [m s$^{-1}$] | 5000 | 5000 | 5000 | - | 1 |
| Shear Wave Velocity [m s$^{-1}$] | 2609 | 2609 | 2609 | - | 1 |
| **10131 – Diorite** | | | | | |
| Bulk Density [g cm$^{-3}$] | 2.35 | 2.89 | 2.72 | 0.10 | 60 |
| Grain Density [g cm$^{-3}$] | 2.65 | 3.08 | 2.82 | 0.07 | 111 |
| Porosity [%] | -0.41 | 23.79 | 2.59 | 3.51 | 60 |
| Thermal Conductivity [W m$^{-1}$ K$^{-1}$] | 0.95 | 3.81 | 2.13 | 0.37 | 241 |
| Thermal Diffusivity [x10$^{-6}$ m$^2$ s$^{-1}$] | 0.53 | 1.73 | 1.05 | 0.17 | 243 |
| Compressive Wave Velocity [m s$^{-1}$] | 4636 | 8167 | 6122 | 737 | 51 |
| Shear Wave Velocity [m s$^{-1}$] | 2532 | 4811 | 3405 | 627 | 51 |
| **10132 – Gabbro** | | | | | |
| Bulk Density [g cm$^{-3}$] | 2.61 | 2.99 | 2.89 | 0.09 | 37 |
| Grain Density [g cm$^{-3}$] | 2.62 | 2.99 | 2.90 | 0.08 | 51 |
| Porosity [%] | -0.46 | 4.55 | 0.62 | 0.92 | 37 |
| Thermal Conductivity [W m$^{-1}$ K$^{-1}$] | 1.54 | 3.07 | 2.18 | 0.23 | 95 |
| Thermal Diffusivity [x10$^{-6}$ m$^2$ s$^{-1}$] | 0.84 | 1.54 | 1.04 | 0.11 | 84 |
| Compressive Wave Velocity [m s$^{-1}$] | 2644 | 7052 | 5025 | 1283 | 27 |
| Shear Wave Velocity [m s$^{-1}$] | 1602 | 3886 | 2820 | 677 | 27 |




**Continuation of Table 3: Petrophysical Properties averaged over the measured samples for each petrographic ID given in the**
**Database Weinert et al. (2020b).**

| 10355 – Micro Diorite | | | | | |
|---|---|---|---|---|---|
| Bulk Density [g cm$^{-3}$] | 2.76 | 2.86 | 2.81 | 0.04 | 13 |
| Grain Density [g cm$^{-3}$] | 2.83 | 2.89 | 2.86 | 0.01 | 13 |
| Porosity [%] | 0.55 | 3.30 | 1.92 | 1.12 | 13 |
| Thermal Conductivity [W m$^{-1}$ K$^{-1}$] | 2.38 | 2.44 | 2.41 | 0.04 | 2 |
| Thermal Diffusivity [x10$^{-6}$ m$^2$ s$^{-1}$] | 1.35 | 1.39 | 1.37 | 0.03 | 2 |
| Compressive Wave Velocity [m s$^{-1}$] | 3566 | 5096 | 4409 | 751 | 5 |
| Shear Wave Velocity [m s$^{-1}$] | 1888 | 2157 | 1998 | 112 | 5 |
| **10210 – Volcanic Rock: Total Average** | | | | | |
| Bulk Density [g cm$^{-3}$] | 2.36 | 3.01 | 2.68 | 0.24 | 8 |
| Grain Density [g cm$^{-3}$] | 2.65 | 2.96 | 2.77 | 0.10 | 10 |
| Porosity [%] | -0.65 | 13.02 | 5.27 | 5.05 | 8 |
| Thermal Conductivity [W m$^{-1}$ K$^{-1}$] | 1.29 | 3.56 | 2.21 | 0.78 | 23 |
| Thermal Diffusivity [x10$^{-6}$ m$^2$ s$^{-1}$] | 0.68 | 2.32 | 1.21 | 0.47 | 23 |
| Compressive Wave Velocity [m s$^{-1}$] | 5845 | 6161 | 6003 | 224 | 2 |
| Shear Wave Velocity [m s$^{-1}$] | 3219 | 3552 | 3386 | 236 | 2 |
| **10224 – Trachyte** | | | | | |
| Bulk Density [g cm$^{-3}$] | - | - | - | - | - |
| Grain Density [g cm$^{-3}$] | - | - | - | - | - |
| Porosity [%] | - | - | - | - | - |
| Thermal Conductivity [W m$^{-1}$ K$^{-1}$] | 1.80 | 1.89 | 1.85 | 0.05 | 3 |
| Thermal Diffusivity [x10$^{-6}$ m$^2$ s$^{-1}$] | 1.05 | 1.08 | 1.07 | 0.02 | 4 |
| Compressive Wave Velocity [m s$^{-1}$] | - | - | - | - | - |
| Shear Wave Velocity [m s$^{-1}$] | - | - | - | - | - |



**Continuation of Table 3: Petrophysical Properties averaged over the measured samples for each petrographic ID given in the Database Weinert et al. (2020b).**

| 10231 – Basalt | | | | | |
|---|---|---|---|---|---|
| Bulk Density [g cm$^{-3}$] | 2.36 | 3.01 | 2.68 | 0.24 | 8 |
| Grain Density [g cm$^{-3}$] | 2.65 | 2.96 | 2.77 | 0.10 | 10 |
| Porosity [%] | -0.65 | 13.02 | 5.27 | 5.05 | 7 |
| Thermal Conductivity [W m$^{-1}$ K$^{-1}$] | 1.30 | 3.45 | 2.07 | 0.78 | 10 |
| Thermal Diffusivity [x10$^{-6}$ m$^2$ s$^{-1}$] | 0.71 | 2.18 | 1.11 | 0.46 | 10 |
| Compressive Wave Velocity [m s$^{-1}$] | 5845 | 6161 | 6003 | 224 | 2 |
| Shear Wave Velocity [m s$^{-1}$] | 3219 | 3552 | 3386 | 236 | 2 |
| **10322 – Dyke Rock: Total Average** | | | | | |
| Bulk Density [g cm$^{-3}$] | 2.65 | 2.78 | 2.65 | 0.07 | 14 |
| Grain Density [g cm$^{-3}$] | 2.61 | 2.96 | 2.83 | 0.12 | 26 |
| Porosity [%] | 0.20 | 14.55 | 9.56 | 3.33 | 14 |
| Thermal Conductivity [W m$^{-1}$ K$^{-1}$] | 1.57 | 3.28 | 2.12 | 0.31 | 47 |
| Thermal Diffusivity [x10$^{-6}$ m$^2$ s$^{-1}$] | 0.79 | 1.67 | 1.07 | 0.15 | 49 |
| Compressive Wave Velocity [m s$^{-1}$] | 1120 | 6706 | 3814 | 1518 | 25 |
| Shear Wave Velocity [m s$^{-1}$] | 673 | 3341 | 2035 | 863 | 25 |
| **10325 – Aplite** | | | | | |
| Bulk Density [g cm$^{-3}$] | 2.78 | 2.78 | 2.78 | - | 1 |
| Grain Density [g cm$^{-3}$] | 2.61 | 2.61 | 2.61 | - | 1 |
| Porosity [%] | 0.20 | 0.20 | 0.20 | - | 1 |
| Thermal Conductivity [W m$^{-1}$ K$^{-1}$] | 2.10 | 3.28 | 2.35 | 0.42 | 7 |
| Thermal Diffusivity [x10$^{-6}$ m$^2$ s$^{-1}$] | 1.09 | 1.67 | 1.24 | 0.21 | 7 |
| Compressive Wave Velocity [m s$^{-1}$] | - | - | - | - | - |
| Shear Wave Velocity [m s$^{-1}$] | - | - | - | - | - |

(c) Author(s) 2020. CC BY 4.0 License.





**Continuation of Table 3: Petrophysical Properties averaged over the measured samples for each petrographic ID given in the Database Weinert et al. (2020b).**

### 10329 – Granitic Subvolcanic Rock

| | | | | | |
|---|---|---|---|---|---|
| Bulk Density [g cm$^{-3}$] | - | - | - | - | - |
| Grain Density [g cm$^{-3}$] | 2.63 | 2.63 | 2.63 | 0.00 | 2 |
| Porosity [%] | - | - | - | - | - |
| Thermal Conductivity [W m$^{-1}$ K$^{-1}$] | 2.56 | 2.61 | 2.58 | 0.03 | 2 |
| Thermal Diffusivity [x10$^{-6}$ m$^2$ s$^{-1}$] | 1.32 | 1.34 | 1.33 | 0.01 | 2 |
| Compressive Wave Velocity [m s$^{-1}$] | 3436 | 3757 | 3596 | 227 | 2 |
| Shear Wave Velocity [m s$^{-1}$] | 2012 | 2069 | 2041 | 40 | 2 |

### 10336 – Porphyric Granite

| | | | | | |
|---|---|---|---|---|---|
| Bulk Density [g cm$^{-3}$] | 2.53 | 2.74 | 2.64 | 0.05 | 13 |
| Grain Density [g cm$^{-3}$] | 2.90 | 2.96 | 2.94 | 0.02 | 13 |
| Porosity [%] | 6.71 | 14.55 | 10.28 | 2.04 | 13 |
| Thermal Conductivity [W m$^{-1}$ K$^{-1}$] | 1.57 | 1.98 | 1.81 | 0.11 | 13 |
| Thermal Diffusivity [x10$^{-6}$ m$^2$ s$^{-1}$] | 0.82 | 1.05 | 0.94 | 0.07 | 13 |
| Compressive Wave Velocity [m s$^{-1}$] | 1120 | 3271 | 1600 | 656 | 13 |
| Shear Wave Velocity [m s$^{-1}$] | 673 | 1874 | 1300 | 298 | 13 |

### 10337 – Porphyric Granodiorite

| | | | | | |
|---|---|---|---|---|---|
| Bulk Density [g cm$^{-3}$] | - | - | - | - | - |
| Grain Density [g cm$^{-3}$] | 2.68 | 2.70 | 2.69 | 0.01 | 4 |
| Porosity [%] | - | - | - | - | - |
| Thermal Conductivity [W m$^{-1}$ K$^{-1}$] | 2.38 | 2.48 | 2.44 | 0.04 | 4 |
| Thermal Diffusivity [x10$^{-6}$ m$^2$ s$^{-1}$] | 1.12 | 1.24 | 1.18 | 0.05 | 4 |
| Compressive Wave Velocity [m s$^{-1}$] | 4993 | 5809 | 5399 | 466 | 4 |
| Shear Wave Velocity [m s$^{-1}$] | 3021 | 3341 | 3192 | 150 | 4 |





**Continuation of Table 3: Petrophysical Properties averaged over the measured samples for each petrographic ID given in the Database Weinert et al. (2020b).**

| 10360 – Porphyric Diorite | | | | | |
|---|---|---|---|---|---|
| Bulk Density [g cm$^{-3}$] | - | - | - | - | - |
| Grain Density [g cm$^{-3}$] | 2.74 | 2.78 | 2.76 | 0.02 | 3 |
| Porosity [%] | - | - | - | - | - |
| Thermal Conductivity [W m$^{-1}$ K$^{-1}$] | 2.05 | 2.56 | 2.18 | 0.17 | 9 |
| Thermal Diffusivity [x10$^{-6}$ m$^2$ s$^{-1}$] | 0.92 | 1.19 | 1.04 | 0.11 | 9 |
| Compressive Wave Velocity [m s$^{-1}$] | 4297 | 5622 | 4857 | 686 | 3 |
| Shear Wave Velocity [m s$^{-1}$] | 2624 | 3310 | 2893 | 366 | 3 |

| 58823 – Diabase | | | | | |
|---|---|---|---|---|---|
| Bulk Density [g cm$^{-3}$] | - | - | - | - | - |
| Grain Density [g cm$^{-3}$] | - | - | - | - | - |
| Porosity [%] | - | - | - | - | - |
| Thermal Conductivity [W m$^{-1}$ K$^{-1}$] | 1.70 | 2.04 | 1.88 | 0.14 | 4 |
| Thermal Diffusivity [x10$^{-6}$ m$^2$ s$^{-1}$] | 0.79 | 1.09 | 1.02 | 0.12 | 6 |
| Compressive Wave Velocity [m s$^{-1}$] | - | - | - | - | - |
| Shear Wave Velocity [m s$^{-1}$] | - | - | - | - | - |

| 58827 – Dioritic Lamprophyre | | | | | |
|---|---|---|---|---|---|
| Bulk Density [g cm$^{-3}$] | - | - | - | - | - |
| Grain Density [g cm$^{-3}$] | - | - | - | - | - |
| Porosity [%] | - | - | - | - | - |
| Thermal Conductivity [W m$^{-1}$ K$^{-1}$] | 2.49 | 2.49 | 2.49 | - | 1 |
| Thermal Diffusivity [x10$^{-6}$ m$^2$ s$^{-1}$] | 1.12 | 1.12 | 1.12 | | 1 |
| Compressive Wave Velocity [m s$^{-1}$] | - | - | - | - | - |
| Shear Wave Velocity [m s$^{-1}$] | - | - | - | - | - |




**Continuation of Table 3: Petrophysical Properties averaged over the measured samples for each petrographic ID given in the Database Weinert et al. (2020b).**

| 58830 – Kersantite | | | | | |
|---|---|---|---|---|---|
| Bulk Density [g cm$^{-3}$] | - | - | - | - | - |
| Grain Density [g cm$^{-3}$] | 2.82 | 2.83 | 2.83 | 0.01 | 3 |
| Porosity [%] | - | - | - | - | - |
| Thermal Conductivity [W m$^{-1}$ K$^{-1}$] | 2.01 | 2.36 | 2.17 | 0.15 | 7 |
| Thermal Diffusivity [x10$^{-6}$ m$^2$ s$^{-1}$] | 0.95 | 1.11 | 1.05 | 0.07 | 7 |
| Compressive Wave Velocity [m s$^{-1}$] | 5431 | 6707 | 6061 | 638 | 3 |
| Shear Wave Velocity [m s$^{-1}$] | 2634 | 3017 | 2815 | 192 | 3 |
| **10907 – Metamorphic Rock (Total)** | | | | | |
| Bulk Density [g cm$^{-3}$] | 2.33 | 3.05 | 2.68 | 0.12 | 281 |
| Grain Density [g cm$^{-3}$] | 2.50 | 3.12 | 2.71 | 0.12 | 336 |
| Porosity [%] | -0.57 | 10.35 | 1.36 | 1.56 | 265 |
| Thermal Conductivity [W m$^{-1}$ K$^{-1}$] | 1.24 | 6.00 | 2.61 | 0.54 | 988 |
| Thermal Diffusivity [x10$^{-6}$ m$^2$ s$^{-1}$] | 0.52 | 5.80 | 1.35 | 0.39 | 902 |
| Compressive Wave Velocity [m s$^{-1}$] | 1553 | 7456 | 4589 | 1069 | 256 |
| Shear Wave Velocity [m s$^{-1}$] | 1103 | 3890 | 2550 | 584 | 256 |
| **10895 – Meta Granite** | | | | | |
| Bulk Density [g cm$^{-3}$] | - | - | - | - | - |
| Grain Density [g cm$^{-3}$] | 2.62 | 2.70 | 2.67 | 0.03 | 9 |
| Porosity [%] | - | - | - | - | - |
| Thermal Conductivity [W m$^{-1}$ K$^{-1}$] | 2.44 | 3.22 | 2.81 | 0.19 | 43 |
| Thermal Diffusivity [x10$^{-6}$ m$^2$ s$^{-1}$] | 1.11 | 1.95 | 1.51 | 0.23 | 43 |
| Compressive Wave Velocity [m s$^{-1}$] | 1093 | 6570 | 4765 | 825 | 25 |
| Shear Wave Velocity [m s$^{-1}$] | 1912 | 3436 | 2809 | 403 | 25 |




**Continuation of Table 3: Petrophysical Properties averaged over the measured samples for each petrographic ID given in the Database Weinert et al. (2020b).**

| 10898 – Meta Gabbro | | | | | |
| --- | --- | --- | --- | --- | --- |
| Bulk Density [g cm$^{-3}$] | - | - | - | - | - |
| Grain Density [g cm$^{-3}$] | - | - | 2.88 | - | 1 |
| Porosity [%] | - | - | - | - | - |
| Thermal Conductivity [W m$^{-1}$ K$^{-1}$] | 2.39 | 2.67 | 2.51 | 0.14 | 3 |
| Thermal Diffusivity [x10$^{-6}$ m$^2$ s$^{-1}$] | 0.95 | 1.11 | 1.05 | 0.09 | 3 |
| Compressive Wave Velocity [m s$^{-1}$] | 5202 | 5789 | 5496 | 415 | 2 |
| Shear Wave Velocity [m s$^{-1}$] | 3111 | 3262 | 3186 | 106 | 2 |
| **10917 – Quartzite** | | | | | |
| Bulk Density [g cm$^{-3}$] | 2.62 | 2.62 | 2.62 | 0.00 | 2 |
| Grain Density [g cm$^{-3}$] | 2.65 | 2.65 | 2.65 | 0.00 | 2 |
| Porosity [%] | 1.16 | 1.16 | 1.16 | 0.00 | 2 |
| Thermal Conductivity [W m$^{-1}$ K$^{-1}$] | 5.24 | 6.11 | 5.73 | 0.44 | 3 |
| Thermal Diffusivity [x10$^{-6}$ m$^2$ s$^{-1}$] | 3.67 | 4.18 | 3.96 | 0.26 | 3 |
| Compressive Wave Velocity [m s$^{-1}$] | - | - | - | - | - |
| Shear Wave Velocity [m s$^{-1}$] | - | - | - | - | - |
| **10926 – Phyllite** | | | | | |
| Bulk Density [g cm$^{-3}$] | - | - | - | - | - |
| Grain Density [g cm$^{-3}$] | - | - | - | - | - |
| Porosity [%] | - | - | - | - | - |
| Thermal Conductivity [W m$^{-1}$ K$^{-1}$] | 2.45 | 4.14 | 3.37 | 0.45 | 11 |
| Thermal Diffusivity [x10$^{-6}$ m$^2$ s$^{-1}$] | - | - | - | - | - |
| Compressive Wave Velocity [m s$^{-1}$] | - | - | - | - | - |
| Shear Wave Velocity [m s$^{-1}$] | - | - | - | - | - |




**Continuation of Table 3: Petrophysical Properties averaged over the measured samples for each petrographic ID given in the Database Weinert et al. (2020b).**

| 10945 – Gneiss | | | | | |
|---|---|---|---|---|---|
| Bulk Density [g cm$^{-3}$] | 2.33 | 2.82 | 2.62 | 0.05 | 186 |
| Grain Density [g cm$^{-3}$] | 2.55 | 2.88 | 2.65 | 0.04 | 199 |
| Porosity [%] | 0.01 | 10.35 | 1.14 | 1.27 | 178 |
| Thermal Conductivity [W m$^{-1}$ K$^{-1}$] | 1.60 | 3.53 | 2.79 | 0.32 | 198 |
| Thermal Diffusivity [x10$^{-6}$ m$^2$ s$^{-1}$] | 0.60 | 1.98 | 1.36 | 0.20 | 194 |
| Compressive Wave Velocity [m s$^{-1}$] | 1886 | 6150 | 4291 | 1062 | 131 |
| Shear Wave Velocity [m s$^{-1}$] | 1103 | 3489 | 2334 | 551 | 131 |
| **10949 – Biotite Gneiss** | | | | | |
| Bulk Density [g cm$^{-3}$] | 2.62 | 2.75 | 2.65 | 0.04 | 11 |
| Grain Density [g cm$^{-3}$] | 2.51 | 2.77 | 2.63 | 0.08 | 13 |
| Porosity [%] | 0.06 | 4.46 | 1.62 | 1.58 | 9 |
| Thermal Conductivity [W m$^{-1}$ K$^{-1}$] | 1.96 | 2.98 | 2.46 | 0.27 | 34 |
| Thermal Diffusivity [x10$^{-6}$ m$^2$ s$^{-1}$] | 0.88 | 1.67 | 1.39 | 0.19 | 30 |
| Compressive Wave Velocity [m s$^{-1}$] | - | - | - | - | - |
| Shear Wave Velocity [m s$^{-1}$] | - | - | - | - | - |
| **10961 – Garnet Biotite Gneiss** | | | | | |
| Bulk Density [g cm$^{-3}$] | 2.73 | 2.74 | 2.74 | 0.00 | 3 |
| Grain Density [g cm$^{-3}$] | 2.74 | 2.76 | 2.75 | 0.01 | 5 |
| Porosity [%] | 0.15 | 0.50 | 0.33 | 0.25 | 2 |
| Thermal Conductivity [W m$^{-1}$ K$^{-1}$] | 2.57 | 4.15 | 3.63 | 0.43 | 18 |
| Thermal Diffusivity [x10$^{-6}$ m$^2$ s$^{-1}$] | 0.71 | 1.94 | 1.64 | 0.29 | 18 |
| Compressive Wave Velocity [m s$^{-1}$] | - | - | - | - | - |
| Shear Wave Velocity [m s$^{-1}$] | - | - | - | - | - |





**Continuation of Table 3: Petrophysical Properties averaged over the measured samples for each petrographic ID given in the Database Weinert et al. (2020b).**

### 10966 – Garnet Plagioclase Gneiss

| | | | | | |
|---|---|---|---|---|---|
| Bulk Density [g cm$^{-3}$] | - | - | - | - | - |
| Grain Density [g cm$^{-3}$] | - | - | - | - | - |
| Porosity [%] | - | - | - | - | - |
| Thermal Conductivity [W m$^{-1}$ K$^{-1}$] | 4.21 | 4.21 | 4.21 | - | 1 |
| Thermal Diffusivity [x10$^{-6}$ m$^2$ s$^{-1}$] | 2.34 | 2.34 | 2.34 | - | 1 |
| Compressive Wave Velocity [m s$^{-1}$] | - | - | - | - | - |
| Shear Wave Velocity [m s$^{-1}$] | - | - | - | - | - |

### 10968 – Hornblende Biotite Gneiss

| | | | | | |
|---|---|---|---|---|---|
| Bulk Density [g cm$^{-3}$] | 2.63 | 2.95 | 2.76 | 0.06 | 15 |
| Grain Density [g cm$^{-3}$] | 2.69 | 2.94 | 2.77 | 0.05 | 29 |
| Porosity [%] | 0.00 | 2.18 | 0.45 | 0.58 | 12 |
| Thermal Conductivity [W m$^{-1}$ K$^{-1}$] | 1.75 | 4.46 | 2.55 | 0.62 | 30 |
| Thermal Diffusivity [x10$^{-6}$ m$^2$ s$^{-1}$] | 0.94 | 2.76 | 1.35 | 0.48 | 25 |
| Compressive Wave Velocity [m s$^{-1}$] | - | - | - | - | - |
| Shear Wave Velocity [m s$^{-1}$] | - | - | - | - | - |

### 10970 – Muscovite Biotite Gneiss

| | | | | | |
|---|---|---|---|---|---|
| Bulk Density [g cm$^{-3}$] | 2.61 | 2.64 | 2.63 | 0.01 | 6 |
| Grain Density [g cm$^{-3}$] | 2.50 | 2.61 | 2.57 | 0.05 | 6 |
| Porosity [%] | 0.33 | 5.35 | 2.70 | 2.11 | 5 |
| Thermal Conductivity [W m$^{-1}$ K$^{-1}$] | 1.92 | 2.62 | 2.36 | 0.23 | 18 |
| Thermal Diffusivity [x10$^{-6}$ m$^2$ s$^{-1}$] | 0.94 | 1.89 | 1.53 | 0.27 | 18 |
| Compressive Wave Velocity [m s$^{-1}$] | - | - | - | - | - |
| Shear Wave Velocity [m s$^{-1}$] | - | - | - | - | - |





**Continuation of Table 3: Petrophysical Properties averaged over the measured samples for each petrographic ID given in the Database Weinert et al. (2020b).**

| 10989 – Amphibolite | | | | | |
|---|---|---|---|---|---|
| Bulk Density [g cm$^{-3}$] | 2.33 | 3.05 | 2.85 | 0.14 | 56 |
| Grain Density [g cm$^{-3}$] | 2.60 | 3.12 | 2.90 | 0.12 | 66 |
| Porosity [%] | -0.57 | 10.35 | 2.16 | 2.14 | 55 |
| Thermal Conductivity [W m$^{-1}$ K$^{-1}$] | 1.24 | 3.80 | 2.20 | 0.40 | 80 |
| Thermal Diffusivity [x10$^{-6}$ m$^2$ s$^{-1}$] | 0.63 | 2.40 | 1.09 | 0.24 | 80 |
| Compressive Wave Velocity [m s$^{-1}$] | 1896 | 6305 | 4931 | 1000 | 19 |
| Shear Wave Velocity [m s$^{-1}$] | 1173 | 3520 | 2806 | 547 | 19 |
| 10990 – Biotite-bearing Amphibolite | | | | | |
| Bulk Density [g cm$^{-3}$] | 2.58 | 2.63 | 2.61 | 0.04 | 2 |
| Grain Density [g cm$^{-3}$] | 2.64 | 2.66 | 2.65 | 0.02 | 2 |
| Porosity [%] | 1.11 | 2.15 | 1.63 | 0.73 | 2 |
| Thermal Conductivity [W m$^{-1}$ K$^{-1}$] | 2.33 | 2.56 | 2.48 | 0.13 | 3 |
| Thermal Diffusivity [x10$^{-6}$ m$^2$ s$^{-1}$] | 1.52 | 1.57 | 1.54 | 0.03 | 3 |
| Compressive Wave Velocity [m s$^{-1}$] | - | - | - | - | - |
| Shear Wave Velocity [m s$^{-1}$] | - | - | - | - | - |
| 10543 – Schist | | | | | |
| Bulk Density [g cm$^{-3}$] | - | - | - | - | - |
| Grain Density [g cm$^{-3}$] | - | - | - | - | - |
| Porosity [%] | - | - | - | - | - |
| Thermal Conductivity [W m$^{-1}$ K$^{-1}$] | 2.00 | 2.03 | 2.01 | 0.02 | 2 |
| Thermal Diffusivity [x10$^{-6}$ m$^2$ s$^{-1}$] | 1.00 | 1.00 | 1.00 | 0.00 | 2 |
| Compressive Wave Velocity [m s$^{-1}$] | - | - | - | - | - |
| Shear Wave Velocity [m s$^{-1}$] | - | - | - | - | - |






**Continuation of Table 3: Petrophysical Properties averaged over the measured samples for each petrographic ID given in the Database Weinert et al. (2020b).**

69620 – Cataclastite

| | | | | | |
|---|---|---|---|---|---|
| Bulk Density [g cm$^{-3}$] | - | - | - | - | - |
| Grain Density [g cm$^{-3}$] | 2.62 | 2.68 | 2.65 | 0.02 | 4 |
| Porosity [%] | - | - | - | - | - |
| Thermal Conductivity [W m$^{-1}$ K$^{-1}$] | 2.65 | 3.39 | 3.00 | 0.26 | 13 |
| Thermal Diffusivity [x10$^{-6}$ m$^2$ s$^{-1}$] | 1.27 | 1.77 | 1.52 | 0.19 | 13 |
| Compressive Wave Velocity [m s$^{-1}$] | 3746 | 5576 | 4540 | 917 | 4 |
| Shear Wave Velocity [m s$^{-1}$] | 1896 | 3388 | 2673 | 703 | 4 |

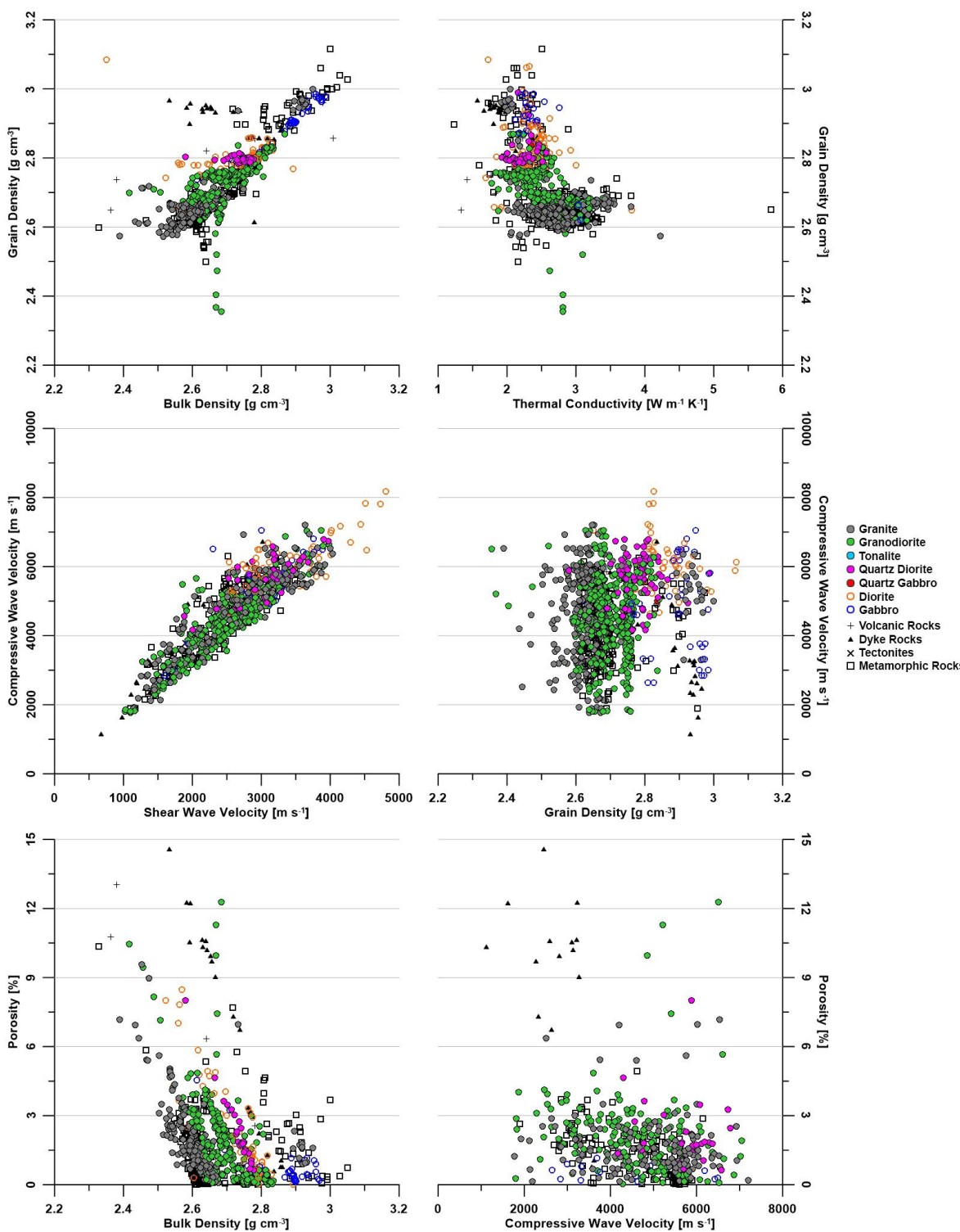

**Figure 4: Scatter plots of the most common petrophysical properties of the data set of Weinert et al. (2020b).**





## 5 Data Availability

The complete dataset of thermophysical and rock mechanical properties can be accessed at https://doi.org/10.25534/tudatalib-278 (Weinert et al. 2020b).

## 6 Discussion

The compilation of comprehensive databases always requires critical review of the input data. Often, published data do not match the minimum requirements of the dataset (as e.g. mentioned in Bär et al., 2020). All data stored in the presented database were measured at the same institution applying the same methods. Therefore, all data presented conform to the same standards and quality requirements. This also ensures a high comparability of the measurements, since the methodological accuracy and errors are identical throughout the dataset. Due to the coherence of the presented dataset, the documented properties are easy

to correlate and can help to understand petrophysical properties within the Mid-German Crystalline High but also allows deriving general correlations between petrophysical properties itself.

Advantageous to many other databases are the metadata attached to each measurement and specimen. Therefore, petrophysical properties can easily be extracted and applied in other studies or for parametrization of local to regional numerical models.

## 8 Sample Availability

Most of the samples are available at the Technical University of Darmstadt and are stored for at least 10 years after finalization of a student's thesis or scientific report. Data that are labelled with 'archive samples of the Institut für Steinkonservierung' (column 'Outcrop Type') were sampled in their archive and it may be possible to request the samples directly at the Institut für Steinkonservierung e. V. In case of well samples, please refer to the Hessian Agency for Nature Conservation, Environment and Geology. For samples of the wells Weiterstadt 1, Worms 3 and Stockstadt 33R please refer to the Bundesverband Erdgas,

Erdöl und Geoenergie e. V.

## 9 Acknowledgement

The authors thank Dr. Karin Kraus from the the Institut für Steinkonservierung e. V. for providing their archive samples for analyzation. Further, the authors are thankful to the Bundesverband Erdgas, Erdöl und Geoenergie e. V. (BVEG) as well as BEB Erdgas und Erdöl GmbG & Co. KG, the owner of the wells, for providing core samples of the crystalline basement for

investigation and publishing. The authors also thank the Hessian Agency for Nature Conservation, Environment and Geology (HLNUG), especially Dr. Johann-Gerhard Fritsche as well as Dr. Gerald Gabriel from the Leibniz Institute for Applied Geophysics (LIAG) and Dr. Sonja Wedmann from the Senkenberg Research Institute and Natural History Museum for providing access to the well cores.





As research assistants Alexander Lambert, Christian Schneider and Stina Krombach helped in measuring the presented data.
The authors are further thankful for the help of Florent Dutheillet de Lamothe within the framework of his internship at Technical University of Darmstadt. Also, the authors are grateful for contributions of Bianca Vogel, Alexander Lambert, Rafael Schäffer, Helmuth Hoffmann, Romain Maire, Alexej Philipp, Johanna Rüther, Jan Niklas Weger, Ulrike Klaeske, Liang Pei, and Hendrik Biewer in the framework of their conducted student theses.

Furthermore, the authors thank for the financial support by the DFG in the framework of the Excellence Initiative, Darmstadt
Graduate School of Excellence Energy Science and Engineering (GSC 1070) as well as the Federal Ministry for Economic Affairs and Energy for financing the present research in the scope of Hessen 3D 2.0 (funding number 0325944A)



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
