# Peer review of "Database of Petrophysical Properties of the Mid-German Crystalline High"

_Earth System Science Data, 2020_

## Referee Comment (RC1) · Anonymous Referee #1 · 4 Sep 2020

Petrophysical rock properties, in particular thermophysical properties, relate to the mineralogy of the distinct rocks. Given the fact that the authors deal with magmatic and metamorphous rocks of the very heterogenous Mid-German Crystalline High, the age ("stratigraphy") is actually less important than the mineralogy of the rock types. This information from thin section analyses should be added. The classification should thus follow the mineralogical characteristics. Additionally, bulk rock geochemical analyses are imperative and should be included. Data from outcrops (weathered) and cores are presented - this point needs to be critically assessed.

———————————————

---

## Referee Comment (RC2) · Anonymous Referee #2 · 5 Oct 2020

Line 41: Can the authors provide a few references here?

Lines 135-136: Surely these volumes should be reported as cm3? Mass per unit volume is a density, no?

Lines 141 and 163: ". . .on the. . ."

Line 182: "oven-dry". At what temperature? The samples were dried in a vacuum oven or a conventional oven/furnace?

Line 187: "Grain density" could be confused with the solid density (i.e. the density of a powdered aliquot of the sample). To avoid confusion, perhaps the authors could write "skeletal" in brackets after "grain"?

Line 192: The authors should make it clear that the "effective porosity" is also, and often, called the "connected porosity".

Lines 206-213: The descriptions of the permeability measurements require more detail, in my opinion. Did the authors use a small confining pressure to ensure that air did not travel along the interface between the sample and the jacket/device? If not, how were the samples clamped to try to prevent this? Did the authors just assume that the Klinkenberg correction was required each time, or did they first determine whether it was indeed necessary? Was the air dried and, if so, how? Scant information is provided about the "mini permeameter". Are the authors talking about a "TinyPerm"? If yes, I'm somewhat sceptical that this device can provide accurate values for low-permeability samples.

Lines 226-227: Although the authors quote a standard practice, I think they should provide more details here. The sentence also requires rewording. For example, what was the force rate used? What was the displacement/strain rate used? How was displacement measured?

Lines 320-232: The descriptions of the triaxial measurements require more detail, in my opinion. What was the displacement/strain rate used? How was displacement measured? What confining fluid was used? How was the confining pressure held constant during the experiment? What type of jacket was used? It also may help to very briefly explain why lower pressures were used for the harder rocks (I assume these samples were too strong to break in the triaxial setup at 30 MPa?). How were the cohesion and the angle of internal friction measured (these parameters were mentioned at the start but not explained)?

Line 233-234: Although the authors quote a standard practice, I think they should provide more details here. The authors are also measuring the indirect tensile strength, not the tensile strength. What stressing rate was used? It might also help to state that the samples were loaded diametrically in compression.

Figure 4: Although choices need to be made, why did the authors choose to plot these graphs? Skeletal density as a function of thermal conductivity? Surely thermal conductivity as a function of porosity would make more sense? What about UCS as a function of porosity? Permeability as a function of porosity? Unless there are good reasons to show these plots, I suggest that the authors rethink what they show here.

Line 358: Is it not useful for the reader to state (with references) whether the trends found are similar to those found in previous studies?

Line 358: Although I understand the importance of a database of laboratory values, these are not the values to be used in large-scale geothermal modelling. For example, laboratory values of permeability underestimate the permeability of a typical rock-mass, which contains fractures and other discontinuities. I think that it's very important for the authors to outline that these measurements require upscaling (so as not to mislead modellers looking for values for their models) and offer a short referenced paragraph that explains how this is typically done.

---

## Author Comment (AC1) · 26 Nov 2020

Dear Referee,

I agree that the mineralogical and geochemical composition of igneous rocks is an important factor and supersedes the importance of stratigraphic age. Metadata of each samples carries both, information on their stratigraphic age (stratigraphic ID) but also their petrographic description (petrographic ID). The petrographic description, if applicable, was determined microscopically on thin sections but also macroscopically on hand pieces of the sampled locations. Data on petrology is described in Chapter 2.1.4, which was extended by the following sentence: "Petrography is either evaluated on thin sections (if applicable) or on fresh hand pieces". Since the focus of the presented study

is set on physical and not chemical properties, I am confident that the provided information on the rock's petrology is sufficient to allow for the classification and interpretation of the physical properties presented. This is especially the case for regional scale studies as presented here, where we intend to define meaningful ranges of the different rock properties for various applications as e.g. geothermal resource assessment studies. Petrophysical properties are furthermore classified and analyzed based on the petrographic description. Table 3 gives average data based on rock type and Figure 4 provides correlations of classified samples. Preparing and analyzing thin sections of all 8,600 samples is both, very expensive and time consuming and can therefore not be included in the study presented. Some of the analyzed samples were furthermore taken from archives and hence, cannot be destroyed for thin section preparation or whole rock geochemistry. Nonetheless, for most of the presented sampling locations, whole rock geochemical analysis is in planning. Since (re-)sampling, sample preparation, measurements (x-ray fluorescence; ICP-MS analysis) and data evaluation is time consuming, additional geochemical information on the sampled outcrops also comprising data from literature references as e.g. Gard et al. (2019) can be added in a second version after publication of the petrophysical properties.

---

## Author Comment (AC2) · 26 Nov 2020

Dear Referee,

thank you for your review on the presented study. Please find my comments point-bypoint as follows:

Line 41: Can the authors provide a few references here?

Thank you for pointing out missing references. Now reads: Nonetheless, even in rather isotropic, homogeneous material such as crystalline rocks, petrophysical properties can vary depending on their geochemical composition and texture (e.g. dataset of Krietsch et al., 2018 and references therein, Weinert et al., 2020a) but also physical appearance, micro-fractures or porosity (e.g. Mielke et al., 2017, Weinert et al., 2020a)

as well as degree of alteration or weathering (e.g. Machek et al., 2013).

Lines 135-136: Surely these volumes should be reported as cm3? Mass per unit volume is a density, no?

Thank you for spotting this typo. Of course, the volume is reported in cm3 and not g cm-3 as it is fixed now.

Changed the preposition to on; thank you.

Line 182: "oven-dry". At what temperature? The samples were dried in a vacuum oven or a conventional oven/furnace?

Added: The samples were dried in a conventional oven at 105 or 60  $^{\circ}$ C (depending on the samples clay content) until constant weight.

Line 187: "Grain density" could be confused with the solid density (i.e. the density of a powdered aliquot of the sample). To avoid confusion, perhaps the authors could write "skeletal" in brackets after "grain"?

Done, thank you for pointing out the possible confusion.

Now states: ... specimen's gas effective (or connective) porosity.
Lines 141 and 163: ". . .on the. . ."

Line 192: The authors should make it clear that the "effective porosity" is also, and often, called the "connected porosity".

Lines 206-213: The descriptions of the permeability measurements require more detail, in my opinion. Did the authors use a small confining pressure to ensure that air did not travel along the interface between the sample and the jacket/device? If not, how were the samples clamped to try to prevent this? Did the authors just assume that the Klinkenberg correction was required each time, or did they first determine whether it was indeed necessary? Was the air dried and, if so, how? Scant information is provided about the "mini permeameter". Are the authors talking about a "TinyPerm"? If yes, I'm somewhat sceptical that this device can provide accurate values for lowpermeability samples.

Samples are mounted and sealed by a rubber sleeve which is pressed against the sample's sidewall with a confining pressure. This is now stated in the revised draft.

The Klinkenberg correction is case dependent. Evaluation of each measurement is done by hand and with consideration of the measurement data.

The "mini permeameter" is combined with the column permeameter and built be a staff member of the university (Hornung & Aigner, 2002 but also described by Filomena et al 2014). Nonetheless, the function is similar to e.g. the TinyPerm 3 of NER Inc. I also share your concerns in accurately measuring low-permeable materials (at least with air drive flow-through permeameters). For the devices used, the detection level is approx. 1E-18 m2. Due to the detection limit, some measurements are only stated as < E-18 m2 and the general count of measurements lack behind other parameters (e.g. wave velocities or thermal properties). In parts, that is the reason for excluding permeability measurements in the presented average data (Table 3) and graphical presentation (Figure 4).

To address your concerns, I added some lines to the methodology section to address this issue. Nonetheless, I think that providing this data is still important and can give indication of the matrix permeability of the recorded samples. The following sentence is

**ESSDD**
added: Detection limit is  $1 \times 10-18 \text{ m}^2$  and it also needs to be addressed, that apparent permeability measurements with the mini permeameter tends to overestimate matrix permeabilities in low-permeable rocks.

Lines 226-227: Although the authors quote a standard practice, I think they should provide more details here. The sentence also requires rewording. For example, what was the force rate used? What was the displacement/strain rate used? How was displacement measured?

The sentence now reads: Unconfined compressive strength was tested in a 1000 kN testing frame (Form+Test Prüfsysteme) according to the ASTM D7012. Tests were both, force and displacement controlled. Strain and force rates were individually set to achieve a testing duration of approx. 10 min. Vertical displacement is measured with an external displacement transducer.

Lines 320-232: The descriptions of the triaxial measurements require more detail, in my opinion. What was the displacement/strain rate used?

Strain rates were individually set for each location/rock strength to meet the testing duration provided by standards. Therefore, nor definite rate can be provided.

How was displacement measured?

The sample strain is constantly logged with an external displacement transducer, measuring the strain of the loading rod against the top platen of the testing frame. Therefore, the deformation of any steel component or compression in hydraulic oil (supplying the main piston) is reduced to a minimum. This information is now added to the revised draft version.
What confining fluid was used? How was the confining pressure held constant during the experiment? What type of jacket was used?

The sample is mounted in a Hoek cell and sealed with a (hard) rubber jacket. Confining pressure is built up and constantly controlled with an external pump. Hydraulic oil is used as the confining fluid. This information is now added to the revised draft version.

It also may help to very briefly explain why lower pressures were used for the harder rocks (I assume these samples were too strong to break in the triaxial setup at 30 MPa?).

Yes, your assumption is correct. I rephrased the concerning sentence to: Due to the rock strength, hard rocks were commonly measured with 5, 10 and 20 MPa confining pressure to achieve sample failure applying the 1000 kN testing frame.

How were the cohesion and the angle of internal friction measured (these parameters were mentioned at the start but not explained)?

To clarify the analysis of both parameters, the following sentence was added: Cohesion and internal friction angle were determined applying the Mohr-Coulomb criterion.

Line 233-234: Although the authors quote a standard practice, I think they should provide more details here. The authors are also measuring the indirect tensile strength, not the tensile strength. What stressing rate was used? It might also help to state that the samples were loaded diametrically in compression.

The section now reads: Tensile (or indirect tensile) strength is determined on rock
disks of 55 and 64 mm diameter at length ratios of approx. 0.5:1 (diameter to length according to ASTM D3967-16.). Therefore, the test procedure follows the Brazilian test procedure, where rock disks are compressed diametrically. Strain rates were individually set to meet the required test duration stated in ASTM D3967-16.

Figure 4: Although choices need to be made, why did the authors choose to plot these graphs? Skeletal density as a function of thermal conductivity? Surely thermal conductivity as a function of porosity would make more sense? What about UCS as a function of porosity? Permeability as a function of porosity? Unless there are good reasons to show these plots, I suggest that the authors rethink what they show here.

Thank you for pointing out your concerns about my choice of the presented figures. Here are some of my thoughts leading to this selection:

1. Due to low to very low porous material, and also very low porosity variability, data will rather plot as a cloud then distinct correlation.

2. Variations in thermal conductivity is not only dependent on porosity but in the case of unweathered intrusive rocks mainly dependent on modal composition and texture, which governs how minerals are distributed and how the contact between minerals/grains is developed.

3. Therefore, thermal conductivity was plotted against skeletal or grain density. Grain density is – in a simplified way – only dependent on the rock chemical or mineralogical compositions – which is why rather distinct clusters are plotted reflecting the differentiation of basic to felsic intrusive rocks.

4. Ratios of compressive and shear wave velocities are used as an indicator of lithology (e.g. Pickett 1963: Acoustic character logs and their applications in formation evaluation, Journal of Petroleum Technology, Vol. 15, No.6) and the ratio often follows statistical equations (e.g. Castagna et al. 1985: Relationships between compressional-wave

ESSDD
and shear-wave velocities in clastic silicate rocks, Geophysics Vol. 50, No.4). Therefore, by plotting compressional vs. shear wave velocity may (but does not necessarily) act as an indicator of measurement quality/accuracy.

Furthermore, the following changes have been done in the reviewed draft:

- 1. The correlation between porosity and thermal conductivity was added
- 2. The correlation of unconfined compressive strength against porosity was added
- 3. The correlation between grain and bulk density was replaced

4. The correlation between grain density and compressional wave velocity was deleted

Line 358: Is it not useful for the reader to state (with references) whether the trends found are similar to those found in previous studies?

Some references to other publications are added and the following paragraph is added to the draft: Found correlation and data ranges are also in range with previous studies of rock of the Mid-German Crystalline High (e.g. Kushnir et al. 2018) but also crystalline basement rocks of various different locations (e.g. Carmichael,1989 and references therein; Mielke et al., 2017; Kushnir et al. 2018) such as e.g. the Gonghe Basin Complex (Weinert et al. 2020a). The correlations of porosity against thermal conductivity but also compressional wave velocity are in accordance with findings of Mielke et al. (2017). Decreasing compressional wave velocities with increasing porosities is in accordance with findings of e.g. Kushnir et al. (2018) or Weinert et al. (2020a). The same negative correlation is found for porosity and unconfined compressive strength (e.g. the presented data set; Li and Aubertin, 2003 Kushnir et al. (2018); Weinert et al., 2020a).

Line 358: Although I understand the importance of a database of laboratory values,
these are not the values to be used in large-scale geothermal modelling. For example, laboratory values of permeability underestimate the permeability of a typical rock-mass, which contains fractures and other discontinuities. I think that it's very important for the authors to outline that these measurements require upscaling (so as not to mislead modellers looking for values for their models) and offer a short referenced paragraph that explains how this is typically done.

Thank you for your concerns and advice in providing more information regarding the usage of such petrophysical properties. In parts, the discussion is rewritten to meet your concerns in this, but also your previous point. The section regarding this comment was extended and now reads: Advantageous to many other databases are the metadata attached to each measurement and specimen. Therefore, petrophysical properties can easily be extracted and applied in other studies or for parametrization of local to regional numerical models. Nonetheless, the presented data may need to be individually processed (e.g. upscaled) to fit application such as large-scaled geothermal modelling. Such data processing can be required since the presented data only represent laboratory-scale matrix properties. Such small-scaled data may vary significantly in larger rock masses. Such scale effects are most prominently observed on matrix and bulk permeabilities. Matrix permeabilities (or permeabilities measured in the laboratory) commonly underestimate the bulk reservoir permeability (as data of e.g. Stober and Jodocy, 2009 or Vidal and Genter, 2018 suggest) due to neglecting permeability of any fractures or fracture network. Datasets such as provided by Achtziger-Zupančič et al. (2017) and Scibek (2020) can provide valuable information for reservoir scale permeability estimations also in nearfields of fault zones. Manning and Ingebritsen (1999), Ingebritsen and Manning (2010) and Stober and Bucher (2015) published data on reservoir scale permeabilities of the upper crust which can be used for assessing and upscaling matrix permeabilities as provided in the presented database. In general, a variety of different upscaling methods are developed, applied, and reviewed (e.g. Wen and Gómez-Hernández 1996; Farmer 2002; Qi and Hesketh, 2005). Techniques can be as simple as applying power law

**ESSDD**
averages on a representative elementary volume (e.g. Qi and Hesketh 2005) but can also require the integration of a fracture network model (e.g. Bao et al., 2012). Since the published methods for upscaling petrophysical properties are plenty, a close consideration of the applied processes is necessary. The method always needs to be chosen regarding the researched application, rock type or geological setting.

Please also note the supplement to this comment: https://essd.copernicus.org/preprints/essd-2020-211/essd-2020-211-AC2supplement.pdf

**ESSDD**

**Supplement:**